# The coordination of anti-phage immunity mechanisms in bacterial cells

Clemente F. Arias [1,2] ✉, Francisco J. Acosta[3], Federica Bertocchini[1], Miguel A. Herrero[4] & Cristina Fernández-Arias [5,6] ✉

Bacterial cells are equipped with a variety of immune strategies to fight bacteriophage infections. Such strategies include unspecific mechanisms directed against any phage infecting the cell, ranging from the identification and cleavage of the viral DNA by restriction nucleases (restriction-modification systems) to the suicidal death of infected host cells (abortive infection, Abi). In addition, CRISPR-Cas systems generate an immune memory that targets specific phages in case of reinfection. However, the timing and coordination of different antiviral systems in bacterial cells are poorly understood. Here, we use simple mathematical models of immune responses in individual bacterial cells to propose that the intracellular dynamics of phage infections are key to addressing these questions. Our models suggest that the rates of viral DNA replication and cleavage inside host cells define functional categories of phages that differ in their susceptibility to bacterial anti-phage mechanisms, which could give raise to alternative phage strategies to escape bacterial immunity. From this viewpoint, the combined action of diverse bacterial defenses would be necessary to reduce the chances of phage immune evasion. The decision of individual infected cells to undergo suicidal cell death or to incorporate new phage sequences into their immune memory would be determined by dynamic interactions between the host's immune mechanisms and the phage DNA. Our work highlights the importance of within-cell dynamics to understand bacterial immunity, and formulates hypotheses that may inspire future research in this area.

The constant threat posed by bacteriophage (phage) infections has driven the development of a wide variety of immune mechanisms in bacteria[1–5]. Among the most common anti-phage defenses are the restriction-modification (RM) systems that detect and attack foreign DNA in the cytoplasm of bacterial cells[6,7]. This immune strategy usually involves two types of enzymes: restriction nucleases that cleave DNA at specific sequences (known as restriction sites), and methyltransferases that modify the same sequences in the host DNA to avoid the autoimmune destruction of self DNA by nucleases[8,9]. The breaks created by restriction nucleases in the viral DNA facilitate its subsequent digestion by other enzymes such as the RecBCD complex[10,11]. The combined action of nucleases and methyltransferases ensures that only unmodified DNA is identified as non-self and destroyed by the cell.

Another defense strategy widespread in bacteria, known as abortive infection (Abi), consists of the suicidal death of infected cells before the completion of the phage replicative cycle, which prevents

[1]CIB, Centro de Investigaciones Biológicas Margarita Salas (CSIC), 28040 Madrid, Spain. [2]Grupo Interdisciplinar de Sistemas Complejos de Madrid (GISC), Madrid, Spain. [3]Departamento de Ecología, Universidad Complutense de Madrid, 28040 Madrid, Spain. [4]Departamento de Análisis Matemático y Matemática Aplicada, Universidad Complutense de Madrid, 28040 Madrid, Spain. [5]Departamento de Inmunología, Facultad de Medicina, Universidad Complutense de Madrid, 28040 Madrid, Spain. [6]Instituto de Medicina Molecular, 1649-028 Lisboa, Portugal. ✉e-mail: tifar@ucm.es; crifer25@ucm.es

the spread of the infection to neighboring bacteria[5,12]. Although this strategy is encoded by an array of different molecular mechanisms, every Abi system requires two complementary functions: one that evaluates the evolution of the infection in the cytoplasm of the infected cell and one that kills the cell when the phage has escaped other bacterial defenses[12]. An example of this logic is the Rex system, one of the first Abi strategies to be described in the literature[13,14]. This mechanism is activated by a sensor protein, RexA, capable of detecting protein-DNA complexes that appear in the bacterial cytoplasm during phage infections. RexA activates a second protein, RexB, that forms ion channels in the cell membrane, inhibiting bacterial growth and leading to the eventual death of the infected cell[15,16].

Abi systems must be tightly controlled so that the suicidal death of the host cell only occurs after the phage has evaded other intracellular defense mechanisms but before it has had time to complete its lytic cycle. Achieving such precise timing is far from trivial. It is usually assumed that RM systems must operate at the early stages of the infection of the bacterial cell and Abi systems in later stages[12,17,18]. However, the mechanisms allowing host cells to unambiguously discriminate early from late phases of phage infections remain largely unexplained.

The previous immune mechanisms must also be coordinated with other intracellular anti-phage defenses. Both RM and Abi systems can be defined as innate: they do not keep a memory of past infections and are unspecific, i.e., they can target any phage infecting the cell. Remarkably, bacterial cells also resort to adaptive immunity: the CRISPR system (present in about 40% of all types of bacteria[19]) creates immune memory against previous phage infections. The number and diversity of known CRISPR systems have steadily grown since the seminal works that led to its discovery[20–24]. Despite their diversity, they all respond to a similar underlying logic. A typical CRISPR immune response consists of three main steps: adaptation, expression, and interference. At the adaptation stage, CRISPR-associated proteins (Cas) bind to the DNA of an infecting phage and cleave a small fragment that is subsequently inserted into the CRISPR array of the host cell, becoming a new spacer. The expression phase occurs during reinfections and consists of the transcription of the CRISPR array into precursor CRISPR RNA, which then undergoes enzymatic cleavage to yield mature CRISPR RNAs that usually contain a single spacer sequence[25]. At the interference stage, these RNAs bind their target nucleotide sequences in the phage genome and recruit Cas nucleases that cleave the phage DNA, thus preventing its replication[26].

The length of the CRISPR array in bacterial genomes ranges from a few dozen to a few hundred spacers[27]. Intuitively, it seems that increasing the number and variety of spacers should make bacterial cells less vulnerable to phages. However, the number of spacers is also subject to an opposite selective force. As noted above, the CRISPR array is usually transcribed as a single unit and then cleaved into individual RNA molecules corresponding to single spacers[24]. This means that all the spacers present in the CRISPR memory of a bacterial cell are recruited in case of infection, even those that cannot target the phage infecting the cell at that moment. An excess of spacers would therefore dilute the number of functional CRISPR units available for the host cell to fight the ongoing reinfection, reducing the overall effectiveness of the system[28]. Furthermore, the predominance of deletions over insertions in bacterial genomes tends to reduce their size[29], which imposes an additional pressure against larger CRISPR arrays.

Given the ubiquity of phages, the fact that some spacers remain in bacterial populations for very long periods of time suggests that not all episodes of infection result in the inclusion of new spacers into the CRISPR library of the infected cells[26,30]. Otherwise, owing to the size constraints discussed above, recent infections would displace earlier ones[31,32], hindering the maintenance of long-term memory. This raises the question of how bacterial cells decide whether or not a given phage or plasmid DNA has to be integrated into their CRISPR arrays. To the best of our knowledge, this issue remains largely unsolved.

This work is motivated by the following questions: How do bacterial cells coordinate the RM, Abi, and CRISPR systems in the course of anti-phage responses? Are these systems redundant or do they target different phage strategies? In what circumstances do bacterial cells incorporate new phages into their CRISPR libraries?

These questions have been addressed at ecological and evolutionary scales in the literature[33–35]. In this work, we take a cell-level perspective. The rationale for this approach is that key phage/bacteria interactions take place in the cytoplasm of infected bacterial cells. It is the outcome of these interactions that determines whether a bacterial cell undergoes suicide or creates new CRISPR spacers against an infecting phage. Therefore, understanding how bacterial immunity operates inside individual cells can provide valuable insight into relevant aspects of phage infections. To address this issue, we use simple mathematical models of the molecular mechanisms that implement anti-phage responses inside individual bacterial cells. These models suggest that the relative rates of phage DNA replication and cleavage suffice to characterize the outcome of a phage infection, i.e., if the host cell survives or it is killed by the phage. These rates allow defining functional categories of phages that differ in their susceptibility to intracellular bacterial immune mechanisms. Within this framework, RM, Abi, and CRISPR systems play complementary roles in the defense of an infected cell by targeting different phage strategies. We show that the coordination between these anti-phage mechanisms in individual cells naturally emerges from the intracellular dynamics of phage/bacteria interactions. Finally, we suggest that those dynamics give rise to a simple molecular mechanism that would allow individual bacterial cells to decide if a new phage must be included in their CRISPR library.

## Results

### The role of restriction nucleases during phage infections

Phages may use two alternative strategies to infect bacterial cells. Lysogenic phages integrate their genome into the bacterial chromosome and replicate without lysing the host. In contrast, lytic phages destroy the host cell to release hundreds of new phages that can infect nearby bacteria[36,37]. Lytic infections, the focus of this work, are usually described by means of qualitative models that represent the key discrete events of the infection (Fig. 1). Although these models are useful to describe the interactions between phages and their hosts, they are insufficient to fully understand the progression of phage infections inside bacterial cells. The fate of the host cell depends on the balance between the effectiveness of its immune mechanisms and the ability of the phage to subvert those mechanisms. However, from the conceptual model shown in Fig. 1 it is impossible to deduce in what circumstances lytic phages prevail over the host's immune defenses or how infected cells decide when they must resort to suicide.

To answer these questions, this qualitative description of nuclease responses has to be translated into quantitative terms. To do that, we will begin by modeling the interactions between nucleases and viral DNA that occur inside infected bacterial cells. We will use the number of phage genomes in the bacterial cytoplasm as an indicator of the progression of the infection. We will assume that if this number reaches a critical value, then the host cell dies; on the contrary, the host cell effectively controls the infection if this number falls below a certain minimum. To model the within-cell dynamics of the infection, we will further assume that phage genomes replicate exponentially and disappear from the host cytoplasm by the enzymatic action of nucleases according to the following equations:

$$\begin{cases} g'(t) = \alpha g(t) - \beta n(t)g(t) \\ g(0) = g_0 \end{cases} \quad \text{for } g_{min} < g(t) < g_{max}, \quad (1)$$

where $g(t)$ and $n(t)$ are the number of phage genomes and of nucleases in the cytoplasm of the host cell at time $t$, respectively. The infection of the bacterial cell is assumed to start at time $t = 0$ with the entry in its

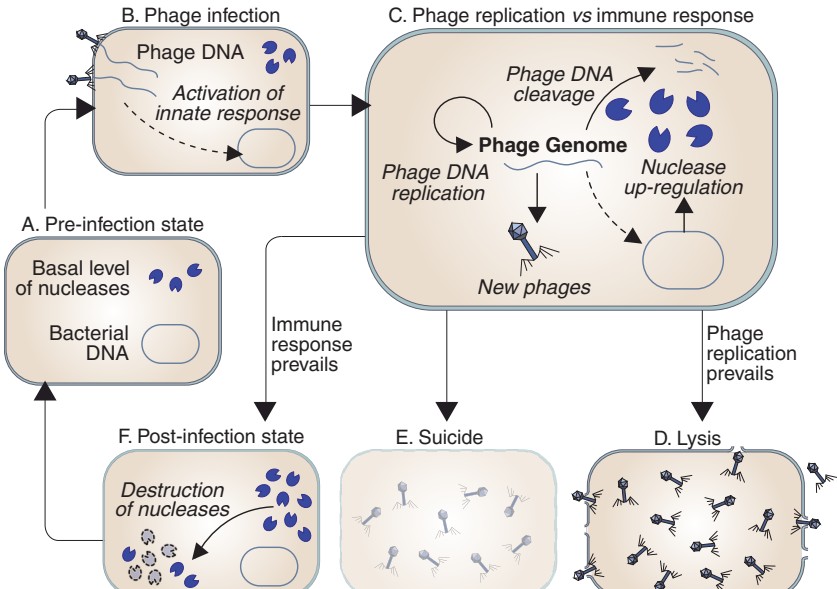

**Fig. 1 | Standard representation of the role of nucleases in lytic infections. A** In the absence of infection, the bacterial cell is free from viral DNA and may display basal levels of restriction nucleases capable of cleaving non-self DNA. **B** In case of infection, the host detects the presence of phage DNA and triggers an innate immune response that includes the upregulation of restriction nucleases. **C** The balance between the replication of the viral DNA and its destruction by bacterial nucleases determines the eventual outcome of the infection. **D** If the phage evades the host's immune response, the bacterial cell dies and releases the newly formed phages to the extracellular space. **E** Alternatively, the infected cell may undergo suicide to prevent the spread of the infection. **F** If the host cell survives, it must deactivate its anti-phage defenses and return to the pre-infection state, which involves the downregulation of the nucleases produced during the infection and also the clearance of the remnants of viral DNA from the cytoplasm.

cytoplasm of $g_0$ phage genomes. Throughout this text, we further assume that there are no superinfections, i.e., that phages prevent the entry of other phages into the host cell once the infection has started[38]. Parameter $g_{min}$ represents the lower threshold that determines phage viability and $g_{max}$ the number of phage genomes that causes the lysis of the host cell. Positive parameters $\alpha$ and $\beta$ represent the phage DNA replication and cleavage rates, respectively.

With regard to nucleases, we remark that their behavior during anti-phage responses is analogous to the clonal expansion and contraction of T cells during adaptive immune responses in vertebrates. The number of nucleases in the bacterial cytoplasm expands when the cell detects the infection and contracts once the infection is resolved (Fig. 1). This analogy is not merely superficial but responds to an identical functional strategy. Both nucleases and effector T cells must remain inactive under normal conditions. Their numbers must rapidly increase (through activation and cell division in the case of T cells and through upregulation in the infected bacterial cell in the case of nucleases) to fight an ongoing infection. After the threat is neutralized, the defenses are de-activated (through apoptosis in T cells and downregulation of nucleases in bacteria[39]), which restores the pre-infection state. T cells and nucleases can be described as elastic systems since they change in size in response to an external stimulus (the infection) and return to their original value when that stimulus disappears. In previous works, we have shown that this feature of T cells can be naturally modeled by means of second-order differential equations (see ref. 40 and ref. 41 for further details on this point). Exploiting the analogy of nucleases with T cells, we will use this approach to model the interaction of nucleases with the phage DNA inside the cytoplasm of an infected bacterial cell as follows:

$$\begin{cases} n''(t) = -\lambda n(t) + \mu g(t) \\ n(0) = n_0 \\ n'(0) = 0 \end{cases} \quad \text{for } n(t) \geq 0, \qquad (2)$$

where $g(t)$ and $n(t)$ are the number of phage genomes and nucleases at time $t$, respectively, $n_0$ is the number of nucleases in the bacterial cytoplasm before the infection, and $\lambda$ and $\mu$ are positive parameters that represent the resistance of nucleases to expand and the force exerted by the phage DNA on the number of bacterial nucleases, respectively[40]. We assume that the number of nucleases in the absence of infection is at a homeostatic equilibrium (hence the condition $n'(0) = 0$).

Putting equations (1) and (2) together, the nuclease response of an individual bacterial cell to a phage infection can be modeled by the following system of differential equations:

$$\begin{cases} n''(t) = -\lambda n(t) + \mu g(t) \\ g'(t) = \alpha g(t) - \beta n(t) g(t) \\ n(0) = n_0 \\ n'(0) = 0 \\ g(0) = g_0 \end{cases} \quad \text{, for } g_{min} < g(t) < g_{max} \text{ and } n(t) \geq 0.$$

(Model 1)

As should be expected, Model 1 captures the elastic nature of nucleases: they are upregulated in the cytoplasm of the host cell in response to the presence of the phage and disappear after the infection is neutralized (see 1 in Fig. 2A). Importantly, this model also reveals key aspects of phage infections that are not evident in its qualitative counterpart (as outlined in Fig. 1). In particular, it suggests that phages may adopt two alternative strategies to evade the nuclease response of a bacterial cell. Phages with very high DNA replication rates prevail by lysing the host cell before the nucleases expand sufficiently to fight the infection (see 2 in Fig. 2A). Less intuitively, phages with very low DNA replication rates give rise to chronic infections of the host cell, characterized by successive cycles of DNA replication and degradation (see 3 in Fig. 2A).

A non-dimensional version of Model 1 allows exploring all the dynamics produced by this model by varying just two parameters: $\alpha$

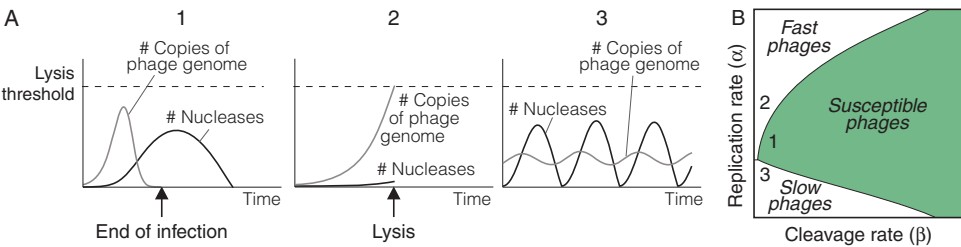

**Fig. 2 | Possible outcomes of the interaction between viral DNA and restriction nucleases in a bacterial cell. A** Model 1 gives rise to three types of outcomes: (1) phages can be successfully eliminated by the expansion of nucleases in the cytoplasm of the infected cell (condition $g(t) < g_{min}$), (2) phages can kill the cell before nucleases have had time to expand (condition $g(t) > g_{max}$), and (3) nucleases and viral DNA can oscillate and give raise to a sustained infection of the host cell (when none of the previous conditions is fulfilled). **B** Model 1 defines an infection space, characterized by the rates of viral DNA replication and cleavage (parameters $\alpha$ and $\beta$ in Model 1, respectively). Within this space, phages can be classified into three functional categories: (1) Phages susceptible to the action of nucleases as the one shown in A1, (2) phages with high replication rates that outrun nucleases (like in A2), and (3) phages with low replication and cleavage rates that induce oscillations in the number of nucleases present in the cytoplasm of the infected bacterial cell (like in A3). Based on the dynamics shown in (**A**), we will label these types of phages as susceptible, fast, and slow, respectively. The details of the simulations are provided in the "Methods" section.

and $\beta$ (see "Methods"). Using this approach, it is possible to define an infection space in which the outcome of anti-phage responses of individual bacterial cells is fully characterized by the rates of phage DNA replication (represented by $\alpha$) and degradation by restriction nucleases (represented by $\beta$). Within this space, phages can be classified into three functional categories that we will label as susceptible (those that can be eliminated by the restriction nucleases of the bacterial cell), fast (those that outrun the expansion of nucleases in the infected cell), and slow (those that cause chronic infections in their host cell) (see Fig. 2B). This classification naturally emerges from a very simple formalization of the standard description of the response of a bacterial cell to a phage infection shown in Fig. 1.

From this perspective, being fast or slow would be alternative phage strategies to evade the immune mechanisms of individual bacterial cells. Although both strategies may lead to the death of the infected cell, their consequences at the scale of bacterial populations would likely be different. Fast phages are capable of rapidly killing their hosts, which probably translates into the high transmission and mortality rates observed in typical lytic infections. The chronic infection of individual cells by slow phages, on the other hand, would result in persistent infections with lower death rates at the scale of the population. This type of infection has been observed in natural populations in which bacteria and phages coexist in a more or less stable equilibrium known as the carrier state life cycle[42]. This state is characterized by persistent infections in which new phages are continuously budded off the host cells or passed down to the progeny of the infected bacterial cells[43]. Within the framework of the infection space, this would be a population-level manifestation of infections by slow phages.

### Bacterial suicide: a strategy against fast phages
In this section, we hypothesize that Abi systems could be optimized to fight fast phages. To support this hypothesis, we will use a mathematical model of the dynamics of typical Abi systems. As discussed above, Abi systems must sense the progression of the infection in the bacterial cytoplasm and kill the cell if the phage cannot be neutralized by other intracellular immune defenses[12]. At the same time, Abi proteins must be tightly suppressed under normal conditions to prevent the death of uninfected bacterial cells[12]. For this reason, the effector mechanisms of Abi systems are normally dormant proteins that only operate when a phage infection is detected in the bacterial cytoplasm[17]. The sensor mechanisms of Abi systems recognize a wide variety of stimuli, ranging from the formation of intermediates of phage replication to the disruption in the expression of host genes caused by the activity of the phage within the infected cell[12]. We will use this last alternative as a model to simulate the behavior of an Abi system with the following features: (i) It monitors the levels of a host protein $s$ whose expression

inside the bacterial cell decreases during the infection; (ii) It triggers the death of the infected cell if this protein disappears from the cytoplasm. The intracellular dynamics of this protein can be simply modeled as follows:

$$\begin{cases} s'(t) = \varphi - \gamma s(t) - \delta g(t) \\ s(0) = \varphi/\gamma \end{cases} \quad \text{for } g_{min} < g(t) < g_{max} \text{ and } s(t) \geq 0, \quad (3)$$

where $\varphi$, $\gamma$, and $\delta$ are positive parameters. This model simulates the dynamics of an intracellular protein that is normally produced at a rate $\varphi$ and disappears exponentially at a rate $\gamma$. These dynamics result in a homeostatic equilibrium given by $s = \varphi/\gamma$, which is taken as the initial condition for $s$. The presence of the phage in the bacterial cytoplasm induces an additional loss of the protein, which is assumed proportional to the amount of phage DNA (parameter $\delta$ is the constant of proportionality). Including equation (3) in Model 1, the simultaneous action of RM and Abi systems in the cytoplasm of an infected bacterial cell can be described by the following system of differential equations:

$$\begin{cases} n''(t) = -\lambda n(t) + \mu g(t) \\ s'(t) = \varphi - \gamma s(t) - \delta g(t) \\ g'(t) = \alpha g(t) - \beta n(t) g(t) \\ n(0) = n_0 \\ n'(0) = 0 \\ s(0) = \varphi/\gamma \\ g(0) = g_0 \end{cases} \quad \text{for } g_{min} < g(t) < g_{max} \text{ , } s(t) \geq 0 \text{ and } n(t) \geq 0.$$

(Model 2)

The three possible results of the phage infection of a bacterial cell shown in Fig. 1 can be explained as alternative outcomes of this model: the phage is neutralized if $g(t) \leq g_{min}$, the host cell is killed by the Abi effector molecules if $s(t) \leq 0$, and the phage survives in the rest of the scenarios. The outcome of the anti-phage response is determined by the first of those conditions to be fulfilled in the course of the infection. Model 2 supports the hypothesis of bacterial suicide as a response to infections by fast phages (Fig. 3A). In particular, this model shows that Abi systems could kill the host cell before these phages can complete their lytic cycles. At the same time, restriction nucleases could neutralize slower phages before the Abi systems have time to induce the death of the host (Fig. 3A). In these cases, the disappearance of the phage from the cytoplasm of the bacterial cell would prevent the activation of the Abi effector mechanisms that kill the host. Whether the bacterial cell undergoes suicide or survives would depend on the dynamics of the enzymes implementing the Abi system relative to the rates of phage DNA replication and destruction.

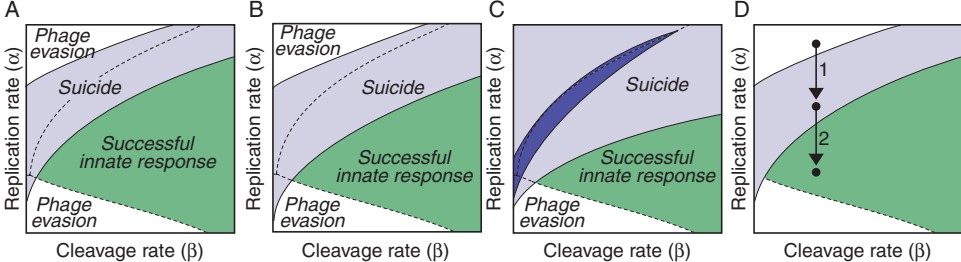

**Fig. 3 | Coordination of innate immune responses in bacterial cells. A** Numerical simulations of Model 2 show that bacterial suicide could take place in a region of the infection space that overlaps with fast phages. This implies that Abi systems could neutralize phages that are not susceptible to restriction nucleases. **B** The suicide region produced by an alternative Abi system (described by equations (4)) is similar to that shown in (**A**). **C** Changing the values of the parameters of equation (3) in Model 2 gives raise to different suicide regions (shown in clear and dark blue,

respectively). **D** A state of dormancy induced by Abi systems in infected cells reduces the rate of phage replication. As a consequence, host cells can resort to suicidal death against phages that would otherwise escape the action of bacterial immunity (1). A sufficiently large reduction in the rate of viral DNA replication by Abi systems may allow for nucleases to eliminate fast phages, preventing both the spread of the infection and the suicide of the host cell (2). The details of the simulations are provided in the "Methods" section.

The previous results support the view of Abi systems as a bacterial strategy to fight fast phages and hint at a new explanation for the role of bacterial suicide during phage infections. Abi systems would not target the same phages as RM systems, and would not be activated when other immune mechanisms fail to control the infection. Quite the opposite, Abi and RM systems would operate simultaneously in the cytoplasm of an infected bacterial cell but they would be programmed to target different categories of phages. RM would neutralize susceptible phages that are beyond the control of Abi systems, whereas the bacterial host cell would undergo suicidal cell death in case of infections by fast phages that restriction nucleases cannot neutralize. The discrimination between susceptible and fast phages would simply emerge from the relative rates at which nucleases and Abi proteins operate inside the infected cell.

In terms of Model 2, RM and Abi systems would occupy different regions of the infection space (see Fig. 3A). We remark that the shape of Abi regions in each individual cell is determined by the dynamics of their sensor mechanism. To clarify this point, let us consider a different Abi system whose sensor mechanism detects the presence of phage DNA. Such a mechanism can be described by the following equations:

$$\begin{cases} s'(t) = \gamma g(t) - \delta s(t) \\ s(0) = 0 \end{cases} \quad \text{for } 0 \leq s(t) \leq s_{max}, \qquad (4)$$

where $s(t)$ and $g(t)$ are the number of sensor proteins and phage genomes in the cytoplasm of an infected bacteria cell at time $t$, respectively. Positive parameters $\gamma$ and $\delta$ now represent the rate at which protein $s$ is produced in response to the phage DNA and the rate of protein degradation, respectively. This model assumes that the concentration of the sensor protein $s$ in an infected bacterial cell increases with the number of genomes in its cytoplasm and decreases exponentially. It further assumes that this protein triggers the death of the host cell whenever its concentration rises above a threshold value $s_{max}$. Substituting equations (3) by equations (4) in Model 2 yields similar results (see Fig. 3B), which shows that Abi systems based on the detection of phage DNA can also target fast phages in bacterial cells.

Regardless of their particular mechanistic details, the range of action of Abi systems is subject to a tradeoff between two alternative constraints. On the one hand, these systems should not kill the host cell if RM systems suffice to control the infection and on the other, they should accelerate the suicide of bacterial cells infected by fast phages. Within the framework of Model 2, these constraints imply that the region occupied by Abi systems in the infection space of bacterial cells should minimize the intersection with susceptible phages while maximizing the one with fast phages. According to our model, this could be achieved by adjusting the kinetic parameters of the sensor enzymes

used by Abi systems (Fig. 3C). This result suggests that natural selection could modulate those parameters to fine-tune the configuration of Abi systems in bacterial cells, which would determine, for instance, what phages should trigger the cell suicide depending on the ecology of each bacterial species.

Abi systems have also been reported to induce a state of dormancy in infected cells by reducing their metabolic rate[5,12,17,44]. This behavior is usually interpreted as an alternative to suicidal death intended to buy time for the action of other immune mechanisms[12]. Our model suggests a different explanation for this phenomenon. Reducing the metabolic activity of infected cells would lower the rate of phage replication, forcing fast phages into the Abi region (Fig. 3D). This would eliminate phages that would otherwise kill the host cell and infect neighboring bacteria (Fig. 3D). Lowering the phage replication rate could also make fast phages susceptible to the action of RM systems, preventing the suicidal death of the infected bacterial cell (Fig. 3D). From this viewpoint, dormancy and suicide would be alternative results of the same mechanism operating against phages with different intracellular replication and destruction rates. Incidentally, this approach would also account for the anti-phage effect of other mechanisms that drastically inhibit phage replication by slowing down the activity of infected cells[45].

## The CRISPR system: a bacterial strategy against slow phages

We have seen that, according to our models, RM and Abi systems would fail to protect bacterial cells against slow phages (Fig. 3). Based on the previous results, we hypothesize that CRISPR systems could be more successful to fight this phage category. Under this hypothesis, CRISPR systems would target phages that are not susceptible to the action of RM and Abi systems. This hypothesis is supported by two main arguments. First, we will show that our mathematical models suggest that CRISPR systems can protect bacterial cells from slow phage infections. Second, we will show that bacterial cells could be able to discriminate susceptible and fast phages from slow phages and selectively incorporate the latter into their CRISPR array. (We will address this issue in the following section.)

To understand the protection conferred on bacterial cells by the CRISPR system, we will model the dynamics of this system in the cytoplasm of infected cells. Since we intend to analyze the effect of CRISPR on the dynamics of the phage DNA inside the host cell, we will not consider the adaptation and expression phases of the CRISPR response. Instead, we will assume that the infected bacterial cell is already equipped with spacers that recognize and attack the DNA of the phage. In this case, the infection activates the interference stage of the CRISPR response in the host cell, which is the focus of our model. At this stage, the action of Cas enzymes is similar to that of restriction

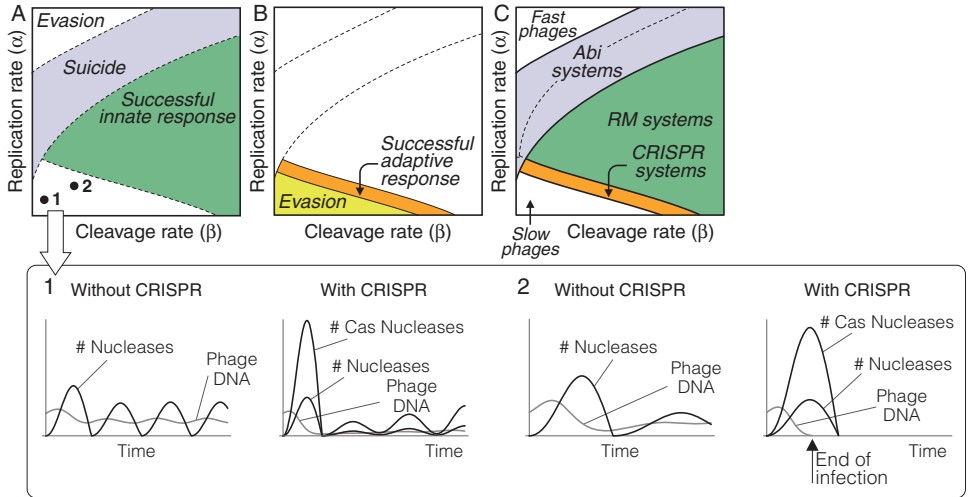

**Fig. 4 | Role of the CRISPR systems in anti-phage responses. A** Within-cell dynamics of slow phage infections in bacterial cells with and without CRISPR systems (as simulated by Model 3 and Model 2, respectively). The activation of CRISPR nucleases in an infected bacterial cell increases the rate of phage DNA degradation. This does not affect some phages that can still cause sustained infections of their host (1). In other cases, the increase in the rate of DNA cleavage due to CRISPR nucleases can entail the end of the infection of the bacterial cell (2). In this case, the CRISPR system could allow bacterial cells to eliminate slow phages that would otherwise escape the control of nucleases. **B** Numerical simulations of Model 3 in the region of the infection space corresponding to slow phages. The CRISPR response creates a new region in the infection space where slow phages can now be neutralized (shown in orange). However, some slow phages can still evade the action of CRISPR spacers and maintain the infection of the cell (yellow region). **C** Regions occupied by different anti-phage immune mechanisms in the infection space of a bacterial cell according to the models used in this work. The details of the simulations are provided in the "Methods" section.

nucleases, since they cleave the phage DNA at specific sites recognized by CRISPR RNAs. As occurs with RM systems, Cas nucleases must expand after the detection of a pathogen, and contract once the phage is eliminated. Therefore, they can also be viewed as elastic systems, which implies that the dynamics of Cas nucleases are conceptually identical to restriction nucleases. For this reason, we will assume that they respond to the same model as RM systems, possibly with different parameters values. Under this assumption, the within-cell dynamics of Cas nucleases can be described by the following equations:

$$\begin{cases} n_c''(t) = -\lambda_c n_c(t) + \mu_c g(t) \\ n_c(0) = 0 \qquad\qquad\qquad \text{for } n_c(t) \geq 0, \\ n_c'(0) = 0 \end{cases} \qquad (5)$$

where $n_c(t)$ and $g(t)$ are the number of nucleases and phage genomes at time $t$, respectively, and $\lambda_c$ and $\mu_c$ are positive parameters (see equations (2)). We assume that Cas nucleases are only expressed in the bacterial cell in case of infection (hence, the initial conditions $n_c(0) = 0$ and $n_c'(0) = 0$).

During infections by phages present in the CRISPR library of the host cell, Cas and restriction nucleases cooperate to cleave the viral DNA and remove it from the cytoplasm. Therefore, the combined action of restriction and CRISPR nucleases on the phage DNA in an infected cell can be modeled by including equations (5) in Model 1:

$$\begin{cases} n''(t) = -\lambda n(t) + \mu g(t) \\ n_c''(t) = -\lambda_c n_c(t) + \mu_c g(t) \\ g'(t) = \alpha g(t) - \beta n(t) g(t) - \beta_c n_c(t) g(t) \\ n(0) = n_0 \qquad\qquad \text{for } g_{min} < g(t) < g_{max}, n_c(t) \geq 0, \text{ and } n(t) \geq 0. \\ n_c(0) = 0 \\ n_c'(0) = 0 \\ g(0) = g_0 \end{cases}$$

(Model 3)

The previous equations simulate a scenario in which the phage is already present in the CRISPR array of the infected bacterial cell. The effect of the CRISPR system on slow phages can be analyzed by applying Model 3 to the particular case of this type of infection (Fig. 4A). Numerical simulations of this scenario show that the immune memory of bacterial cells could neutralize infections by slow phages that are resistant to restriction nucleases (Fig. 4B). As noted above, this result supports the hypothesis that the CRISPR system could constitute a bacterial strategy against slow phages.

In summary, our models suggest that fast and slow phages, capable of evading the action of restriction nucleases, could be targeted by Abi and CRISPR systems, respectively (Fig. 4C). We have seen that the kinetic parameters of the enzymes that implement Abi systems in bacterial cells could account for the selective action of these systems on fast phages. In the next section we hypothesize a mechanism that could explain how could bacterial cells identify slow phages and selectively incorporate them into their CRISPR arrays.

## How do bacteria decide to include new phages in the CRISPR array?

The creation of new CRISPR spacers in a bacterial cell requires the formation of stable complexes between Cas proteins Cas1 and Cas2 during the adaptation stage of the CRISPR immune response[25]. Cas1-Cas2 complexes then interact in the cytoplasm of the infected cell with the fragments of phage DNA that will be added to the CRISPR library[46]. This interaction obviously requires that fragments of phage DNA are still present in the host cytoplasm after the formation of Cas1-Cas2 complexes. Based on this observation, we hypothesize that the incorporation of new spacers into the CRISPR array of a bacterial cell is determined by the permanence of phage DNA in its cytoplasm (as already suggested in ref. 47). This mechanism relies on two basic assumptions. The first one is that the formation of Cas1-Cas2 complexes is delayed with respect to the initiation of the innate anti-phage responses in the infected bacterial cell, which implies that when these complexes appear in the cytoplasm the phage DNA is already undergoing nuclease-mediated degradation. Our second assumption is that Cas1-Cas2 complexes can only create new spacers if they encounter suitable fragments of phage DNA in the host cytoplasm (see Fig. 5). Both assumptions are supported by the observation that Cas1-Cas2 complexes create new spacers using the debris that appears from the degradation of the viral DNA by nucleases[25].

Model 2 suggests that the time needed by nucleases to remove susceptible phages from the cytoplasm of an infected bacterial cell is not homogeneous. Phages with greater rates of DNA cleavage and replication take less time to disappear from the host's cytoplasm (Fig. 6A). In contrast, the DNA from slow phages remains in the cytoplasm of an infected bacterial host cell for longer periods of time (see Figs. 6A and 2A). In consequence, only slow phages would verify the second assumption of the model outlined in Fig. 5. Therefore, the delay

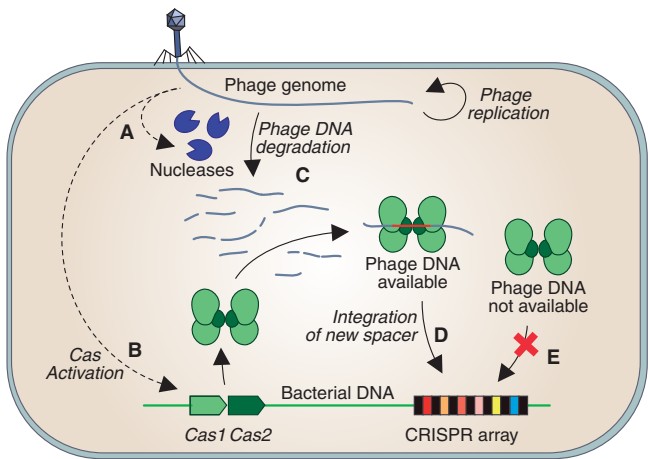

**Fig. 5 | Hypothesized model of the inclusion of new phages in the CRISPR memory of a bacterial cell.** The detection of a phage infection in a bacterial cell up-regulates restriction nucleases that cleave the viral DNA at specific restriction sites, fostering its subsequent digestion by other bacterial enzymes (**A**). Our model assumes that Cas enzymes are activated in the late stages of the innate response of the bacterial host to the infection (**B**), once the viral DNA has already undergone a certain degree of nuclease-mediated degradation (**C**). New spacers against the infecting phage can only be created by the host cell if fragments of its DNA are still present in its cytoplasm after the formation of Cas1-Cas2 complexes (**D**). Otherwise, Cas nucleases have no substrate to act, which prevents the incorporation of the phage into the CRISPR array of the bacterial cell (**E**).

of Cas activation relative to the velocity of action of restriction nucleases would allow infected bacterial cells to discriminate between slow and fast phages and to exclude the latter from their CRISPR library. This model is supported by the observation that lower replication rates caused by the presence of replication-defective phages facilitate the acquisition of new spacers by bacterial cells during phage infections[48]. We stress that this does not imply that the immune memory is effective against all slow phages. Model 3 shows that the CRISPR spacers may not be prevent chronic infections of the bacterial cell in some cases (light gray region in Fig. 6B).

This hypothetical model has interesting functional implications regarding the behavior of the CRISPR systems during reinfections. Should bacterial cells add new spacers against phages that are already represented in their CRISPR array in case of reinfection? This issue concerns an important aspect of the bacterial immune memory that critically affects the function of CRISPR systems. Apparently, creating new spacers against the same phage during each episode of reinfection would reduce the utility of the system by decreasing the diversity of spacers in the CRISPR library of the bacterial cell. This would be especially so in the case of phages that cause frequent reinfections, which would soon displace other phages and dominate the bacterial immune memory[49]. We will next show that the mechanism of immune formation hypothesized above provides a straightforward solution to this issue.

Let us consider two different scenarios of reinfection. First, let us suppose that the rate of phage DNA cleavage has not changed since previous infections by the same phage. Model 3 shows that, in this case, the immune memory of the bacterial cell suffices to eliminate the phage (red circle in Fig. 6C). Let us now consider a second scenario in which reinfecting phages already present in the CRISPR library are less susceptible to the host's nucleases than in previous infections (owing, for instance, to modifications in the restriction sites of their DNA or to mutations in the sequence recognized by the CRISPR spacers[47,50]). The effects of a reduced rate of viral DNA degradation by nucleases can be simulated by lowering the value of parameter $\beta$ in Model 3. Within the framework of the infection space, this entails the displacement of the model results to the left, the magnitude of this displacement being

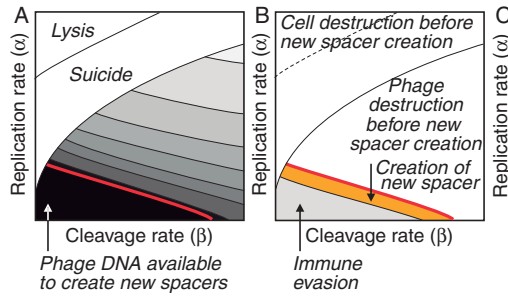
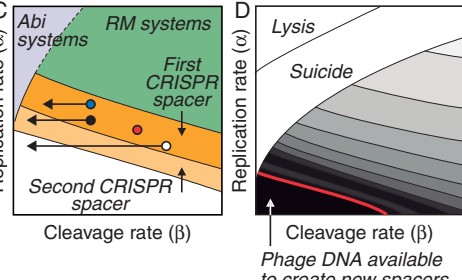

**Fig. 6 | Functional consequences of CRISPR. A** Contour plot of the permanence of phage DNA in the cytoplasm of infected bacteria according to Model 2. Different shades of gray indicate the time needed for nucleases to remove the phage DNA from the cytoplasm of an infected bacterial cell (lighter colors indicate a faster elimination of phage DNA) Higher rates of DNA replication and cleavage reduce the time needed by nucleases to eliminate the phage DNA. The black region corresponds to slow phages that evade the action of nucleases. We hypothesize that the time it takes for nucleases to remove phage DNA from the cytoplasm of an infected bacterial cell provides a mechanistic criterion to include new phages in the CRISPR library. The late activation of the Cas enzymes (indicated by the red line) would constrain the creation of new spacers against susceptible phages as shown in B. The phage DNA would only be available for these enzymes in the region that corresponds to slow phages. **B** Some susceptible phages would be eliminated from the cytoplasm of the host cell before the formation of Ca1-Cas2 complexes (indicated by the red line). Fast phages, on the other hand, would kill the cell before the activation of the Cas enzymes. During infections by slow phages, Cas1-Cas2 would

have enough time to create new spacers and therefore include the infecting phage in the CRISPR array of the infected cell (orange region). The creation of CRISPR spacers against very slow phages would not suffice to control the infection (gray region). **C** Reinfections of bacterial cells by phages that have changed their rates of DNA cleavage have three possible outcomes: the host cell can eliminate the phage (blue circle), it can create a new spacer against the same phage (black circle), it can be killed by the phage (white circle). If the rate of phage DNA destruction has not changed, then the immune memory of the bacterial cell suffices to neutralize the reinfection (red circle). **D** Contour plot of the permanence of phage DNA in the cytoplasm of infected bacteria according to Model 3. Lighter shades of gray indicate shorter permanence time of the phage DNA in the cytoplasm of the bacterial cell. The red line indicates the time of Cas enzymes activation. The action of CRISPR nucleases accelerates the degradation of the phage DNA (see (**A**) for comparison). This implies that the Cas enzymes cannot create new spacers in case of reinfection unless the rate of phage DNA has decreased since previous infections. The details of the simulations are provided in the "Methods" section.

proportional to the reduction in the rate of DNA destruction. This scenario has three possible outcomes. A first alternative is that the reduction in the rate of DNA cleavage does not prevent the neutralization of the phage by the bacterial cell (blue circle in Fig. 6C). In other cases, the infected cell could create new spacers against the phage. This could raise the rate of destruction of the viral DNA and terminate the infection (black circle in Fig. 6C). The creation of new spacers would not ensure the protection of the host cell against all reinfecting phages. Reducing the rate of DNA destruction might allow the phage to evade the action of restriction and CRISPR nucleases and maintain the infection (white circle in Fig. 6C).

From the previous results it follows that creating new spacers against phages already present in the CRISPR array of a bacterial cell could be superfluous or advantageous depending on the circumstances. New spacers would be useful against phages that can evade the immune memory of the host cell (black circle in Fig. 6C) but not against phages that can be successfully eliminated thanks to the CRISPR spacers already in place during reinfections (blue and red circles in Fig. 6C). Can bacterial cells discriminate between these situations? The same model that accounts for the creation of spacers during the first encounter between a bacterial cell and a phage could provide an answer to this question. Using Model 3 it is possible to simulate the effect of the CRISPR memory on the duration of the phage DNA in the cytoplasm of the host cell during reinfections (Fig. 6D). This model suggests that the DNA of phages that can be controlled by the host's immune memory disappears rapidly from the bacterial cytoplasm. According to the hypothetical mechanism presented above, the absence of phage DNA would prevent the action of the Cas enzymes and, consequently, the creation of new spacers (Fig. 6D). In contrast, a reduced rate of phage DNA cleavage would lengthen the permanence of the phage DNA in the host cell, allowing Cas enzymes to create new spacers for the same phage (Fig. 6D). This model would account for the increased acquisition of new spacers during reinfections by phages that have undergone mutations in the targeted DNA sequences as compared to phages without mutations[47]. According to our model, this would be the expected result of lower cleavage rates caused by the reduced affinity of Cas enzymes for the phage DNA.

In summary, the mechanism of immune memory formation in bacterial cells postulated above accounts for the decision to incorporate a phage into the CRISPR array the first time it infects a host cell. The same model provides a functional criterion to include new spacers against phages already present in the CRISPR library during reinfections. Finally, it explains how could CRISPR systems target slow phages, which are precisely the type of infections that cannot be neutralized by innate immune mechanisms.

## Discussion

In this work, we use a simple mathematical description of the standard models of phage/bacteria to gain insight into the within-cell dynamics of phage infections. This approach reveals unexpected features of anti-phage defenses that could hardly be understood from their qualitative descriptions, and suggests testable hypotheses about the coordination of immune mechanisms in individual bacterial cells:

- Phages with high and low DNA replication rates could avoid the action of restriction nucleases in the cytoplasm of infected cells. We have labeled these phages as fast and slow, respectively. Whereas fast phages would give rise to typical lytic cycles, slow phages could lead to the persistent infections observed in carrier state life cycles.
- Abi and CRISPR systems may be more successful in targeting fast and slow phages, respectively.
- The decision of bacterial cells to create new spacers would be determined by the permanence of the phage DNA in the cytoplasm of the cell and the timing of activation of the Cas enzymes.

As a consequence, the creation of new spacers in the CRISPR may be biassed towards slow phages.
- During reinfections, bacterial cells may create new spacers against phages that are already present in their CRISPR library if the rate of phage DNA destruction has decreased since previous infections (owing to modifications in the restriction sites recognized by nucleases or to mutations in the sequences targeted by the spacers present in the CRISPR array of the host cell).

The mathematical models presented in this work support the plausibility of these hypotheses but their validation requires an empirical approach. In this regard, techniques that allow monitoring single-cell dynamics[51] could be used to experimentally test if different infection strategies (e.g., lytic or chronic) respond to different rates of phage DNA replication and cleavage. Alternatively, manipulating those rates would lead to predictable changes in the outcome of phage infections. For instance, it would be theoretically possible to shift between the carrier state and the lytic life cycles by artificially decreasing or increasing the rate at which the DNA of a given phage replicates inside bacterial cells.

Our models show that within-cell dynamics of anti-phage systems could account for key aspects of bacterial immunity. The importance of dynamics is explicitly acknowledged in the explanation of other bacterial mechanisms. For instance, it is widely accepted that the function of toxin-antitoxin systems depends on the different rates of degradation of two molecules, a toxin that can kill the host cell and an antitoxin that protects the host from this effect[52]. If protein synthesis is halted (owing to nutritional or environmental stress[53]), the antitoxin disappears from the bacterial cytoplasm before the toxin, which has a lower degradation rate[54]. In these circumstances, the toxin can no longer be neutralized and the host cell dies. The differences in the rates of degradation of the toxin and the antitoxin are obviously crucial to understanding the logic of this mechanism.

Our results suggest that taking into account the rates of activity of the molecules involved in anti-phage defenses could also be necessary to understand the coordination of anti-phage defenses inside bacterial cells. This approach sets the ground for a new framework for bacterial immunity in which phages and bacterial cells can be viewed as competing to occupy an infection space. Within this space, characterized by the rates of phage replication and destruction, phages would exploit the regions that are beyond the reach of bacterial defenses. In their turn, RM, Abi, and CRISPR systems would be designed to minimize the extent of those regions. The configuration of the infection space presented in this work does not intend to be an accurate, universal description of bacterial infection spaces. For one thing, it corresponds to bacterial cells equipped with RM, Abi, and CRISPR systems, a situation that is not a universal norm among bacterial species. Moreover, the shape of the regions occupied by the Abi, and CRISPR responses in this space depends on the particular mathematical formalization used to model their underlying molecular mechanisms. Instead, the infection space is intended as a useful abstraction of phage/bacteria interactions whose main utility derives from its capacity to provide valuable insight into the within-cell dynamics of phage infections. The interest of this approach transcends the particular situations considered in our models. For instance, bacterial cells that only use RM and Abi systems would still be susceptible to slow phages, whereas the combination of RM and CRISPR responses alone would not suffice to protect the host against fast phages.

The infection space, as defined in this work, is a cell-scale feature: it determines the possible fates of an infected bacterial cell (i.e., whether or not it survives the infection) as a function of the rates of phage DNA replication and destruction. However, this concept could also be formalized at the scale of bacterial populations. Considering the population as a collection of individual cells, it can be characterized by the set of its infection spaces. Variations in the metabolic status or in

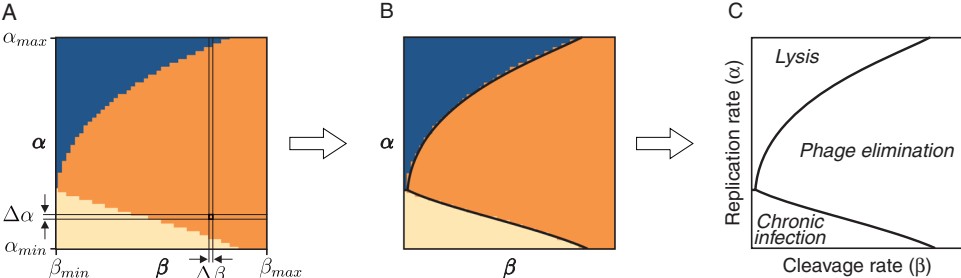

**Fig. 7 | Characterization of the infection space from the simulations of Model 1.** **A** Numerical simulations of Model 1 have three possible outcomes: cell lysis (blue), control of the infection (orange), and chronic infection (light yellow). The simulations were performed for a range of values of parameters $\alpha$ and $\beta$, from a minimum ($\alpha_{min}$ and $\beta_{min}$, respectively) to a maximum value ($\alpha_{max}$ and $\beta_{max}$, respectively) with steps $\Delta\alpha$ and $\Delta\beta$, respectively. **B** These simulations give rise to a discrete version of the infection space. **C** For the sake of simplicity, the figures show a representation of the boundaries between the different regions of the infection space. The rest of the figures in the text were constructed with the same logic.

the anti-phages implemented in each cell imply that the individual spaces within a population are not necessarily identical. The heterogeneity of the population in terms of its infection spaces is also affected by ecological and evolutionary events taking place at different time scales. This diversity plays a key role in the dynamics of the population during phage infections. For instance, some phages might be fast to some cells of the population but susceptible to others, which would entail differential mortality rates among the cell-level configurations of the infection space coexisting in the population. In turn, this would give rise to dynamic changes in the relative abundance of infection spaces in the course of phage infections. This view of the dynamics of bacterial populations as emerging from the structure of individual infection spaces suggests a promising bottom-up strategy to link intracellular and ecological aspects of phage/bacteria interactions.

The concept of an infection space also admits an alternative, static formalization at a population scale. From a population viewpoint, susceptible phages can be defined as those that are susceptible to all the cells of the population. From this definition, the region of the population-level infection space occupied by susceptible phages is given by the intersection of the corresponding regions in the cell-level infection spaces contained in the population. An analogous argument can be used to define phages as slow or fast at the scale of the population. The heterogeneity of individual-level infection spaces implies a greater complexity for the population-level space since new regions appear where phages may be susceptible to some bacterial cells but not for others. However, even a simple setting of the population-level space allows exploring interesting evolutionary consequences of immune mechanisms that operate within the cytoplasm of individual bacterial cells. For instance, it is clear that phages should avoid the regions of the population-level infection space where they can be neutralized by all the cells of the population (such phages would be unable to complete their life cycle and progress with the infection). A possible phage strategy to achieve that end, widely documented in the literature[47,55], consists in modifying the restriction sites or the sequences targeted by CRISPR systems in the viral DNA, which reduces the likelihood of nuclease-mediated recognition and cleavage and, consequently, lowers the rate of phage destruction. Within the framework of the infection space, this strategy would allow phages to escape from the susceptible regions. Our models suggest the existence of a less obvious selective pressure on phages. By increasing or decreasing their rates of proliferation inside their bacterial hosts, phages could also avoid immune destruction by evading the action of bacterial nucleases. From this viewpoint, the Abi and CRISPR systems could be considered as adaptations to counteract these phage strategies.

The concept of an infection space naturally emerges from the within-cell dynamics of well-known immune mechanisms that operate inside infected bacterial cells. Very simple mathematical descriptions

of these mechanisms illustrate the explicative power of the dynamic aspects of anti-phage defenses. We believe our approach provides a fresh perspective to interpret the interactions between phages and bacterial cells and paves the way to a better understanding of bacterial immunity.

## Methods

The numerical simulations of the models were performed with Wolfram Mathematica®. To elaborate the figures of the infection space, the models were run for a discrete set of values of parameters $\alpha$ and $\beta$. For each set of parameters, the outcome of the simulations of Model 1 was determined by the following conditions: (1) if $g(t) \leq g_{min}$, the infection is controlled, (2) if $g(t) \geq g_{max}$, the host cell is lysed, and (3) otherwise the infection is chronic, which may entail the eventual death of the host cell. The outcome of the simulations of Model 2 was subject to an additional condition, imposed by the Abi sensors: if $s(t) \geq s_{max}$, the infected cell undergoes suicidal death. Based on the results of each simulation, the infection space was qualitatively described as shown in Fig. 7.

We used a non-dimensional form of Model 2 to explore its behavior in a wide range of parameter values:

$$\begin{cases} n''(t) = -n(t) + g(t) \\ g'(t) = \alpha^* g(t) - \beta^* n(t)g(t) \\ n(0) = 0 \\ n'(0) = 0 \\ g(0) = 1 \end{cases} \quad \text{for } g_{min}^* < g(t) < g_{max}, \text{ and } n(t) \geq 0,$$

(6)

where: $\alpha^* = \frac{\alpha}{\sqrt{\lambda}}$, $\beta^* = \frac{\beta\mu g_0}{\lambda\sqrt{\lambda}}$, and $g_{min}^* = \frac{g_{min}}{g_0}$. For the sake of simplicity, we use the notation $\alpha^* = \alpha$, $\beta^* = \beta$, and $g_{min}^* = g_{min}$.

For Model 2 and Model 3 we chose suitable values to show that Abi and CRISPR systems can be effective against fast and slow phages, respectively.

The specific values shown in the figures were the following:

- Figure 2. (B) Numerical simulations of Model 1 for $\lambda = 1$, $\mu = 1$, $n_0 = 0$, $n'(0) = 0$ $g_0 = 1$, $g_{min} = 10^{-6}$, $g_{max} = 70$, $0.01 \leq \alpha \leq 11$, $0.2 \leq \beta \leq 10$, $\Delta\alpha = 0.2$, and $\Delta\beta = 0.2$. (A) As in B, with $\alpha = 4$, $\beta = 1$ (susceptible phage), $\alpha = 5$, $\beta = 0.5$ (fast phage), and $\alpha = 0.3$, $\beta = 0.2$ (slow phage).
- Figure 3. (A) Numerical simulations of Model 2 with: $\varphi = 50$, $\gamma = 2\delta = 4$. The rest of the parameters are the same as in Fig. 2. (B) Model 2, using equations (4) instead of equations (3) to simulate the dynamics of Abi sensors, with $\gamma = 2.2$, $\delta = -0.2$, and $= s_{max} = 20$. (C) Numerical simulations of Model 2 with: $\varphi = 50$, $\gamma = 5$, and $\delta = 2$ (light gray region) and with $\varphi = 30$, $\gamma = 1.5$, and $\delta = 0.5$ (dark gray region). (D) Same as in A.

- Figure 4. (A) Same as Fig. 2. (B–D) Numerical simulations of Model 3 for slow phages with $\lambda_c = 1$ and $\mu_c = 3$. The rest of the parameters are the same as in Fig. 3A.
- Figure 6. (A) Same as in Fig. 2. (B–D) Same as in Fig. 4B.

## Reporting summary

Further information on research design is available in the Nature Portfolio Reporting Summary linked to this article.

## Data availability

No datasets were generated or analyzed during the current study.

## Code availability

The code used to analyze the models presented in this work is available at the Notebook Archive (https://notebookarchive.org/2022-08-dbzqzpq).

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

## Acknowledgements

F.B. and C.F.A. are grateful to the Roechling Foundation for its support. C.F.-A. was partially supported by the FCT grant no. EXPL/BIA-BIO-0644/2021.

## Author contributions

Conceptualization: C.F.A. and C.F.-A. Mathematical modeling: C.F.A. Methodology: C.F.A., F.J.A, F.B., M.A.H., and C.F.-A. Investigations: C.F.A., F.J.A., F.B., M.A.H., and C.F.-A. Funding acquisition: C.F.-A. Writing—original draft: C.F.A. and M.A.H. Writing—review & editing: C.F.A., F.J.A., F.B., M.A.H., and C.F.-A.

## Competing interests

The authors declare no competing interests.
