## [Peer Review File · Nature Communications]

The coordination of anti-phage immunity mechanisms in bacterial cellsReviewer #1 (Remarks to the Author):

Arias et al. propose an interesting, and likely useful, formalism for understanding the outcome of phage infection in a cell with multiple defense systems. In their view, the defense strategy the host uses will ultimately depend on the replication rate of the phages inside the cell, and different regulatory circuits and outcomes for different types of defense necessarily lead to these distinct outcomes. They make this formalism explicit with a series of mathematical models of progressing complexity. I have no issue with the construction of the models the authors present, which is very clear and intuitive, but ultimately the simplicity of the formalism seriously constrains its usefulness in making inferences about bacteria-phage ecological and coevolutionary dynamics.

Overall, this manuscript reads much more like a "Perspectives" type article than a research article and may be more appropriate for submission under that category.

Major Comments:

1. These within-cell dynamics are never embedded in a population model, seriously restricting inferences about how these defense systems will interact in even a simple 1 host – 1 phage setting. This absence is felt particularly in conversations of Abi, given that Abi is a defense that only pays a fitness payoff at a population scale.
2. Many evolutionary arguments are made throughout the paper, but the division of state space between immune systems seen is entirely a result of the assumed dynamics of different kinds of defense, not an evolved partitioning of that space. To make any evolutionary arguments the authors should formalize those arguments mathematically (e.g., by looking at the fitnesses of competing strains in the model, perhaps taking a game-theoretic approach)
3. The argument made in the section "How do bacteria decide to include new phages in the CRISPR array?" is entirely verbal, and has no mathematical analysis. I suggest the authors formalize this intuition in equations. This entire section is largely speculation. In particular, the last paragraph of this section makes a number of claims, none of which I think are actually addressed by the model the authors present (the decision to incorporate a spacer and what mechanisms make that decision, why CRISPR systems would target slow phage – see next section)
4. The treatment of CRISPR systems in the paper, while a central point of the work (see title), is extremely simplistic, and I would argue not particularly helpful in terms of understanding these systems. CRISPR systems are implemented in Model 3 identically to an RM system with slightly different parameters – in fact one could think of the same model as representing a cell with an Abi system and two RM systems with different regulatory and kinetic parameters. Nothing is said, really, about adaptive immunity in these models. Also, importantly, there is no real justification for why CRISPR systems would have parameters that would lead them to better target slow phage. The last paragraph of this section makes a number of evolutionary arguments that are not backed up or formalized at all – and may not hold at all in a population setting.
5. The priming phenomenon common to most CRISPR systems (rapid uptake of spacers against phages the host already has immunity towards) is not mentioned in the paper, and seriously complicates the arguments about spacer acquisition decisions made by the authors (though see <https://doi.org/10.1073/pnas.1602639113> for information that may be useful/supportive of the argument). Similarly, while the authors correctly point out that spacers can be acquired from the byproducts of RM defense, defective phage can also facilitate spacer uptake (<https://doi.org/10.1038/ncomms5399>).
6. The CRISPR modeling literature is extremely rich, grapples with many of the questions discussed here, and largely ignored in this manuscript (with a few exceptions). For example, <https://doi.org/10.1371/journal.pcbi.1004603> seems particularly relevant.
7. There is no exploration of parameter space nor justification of the parameters used in the manuscript, making it hard to judge the generality of the results shown.

Minor Comments:

1. Please make all code for analyses publicly available (e.g., on github)
2. It seems like many of these models should be amenable to mathematical analysis, not just numerical approaches, and such solutions would be quite useful
3. The line "This selective force drives frequent mutations..." is incorrect – depletion of restriction sites does not drive mutation. The word "mutations" should be replaced by "substitutions"

Reviewer #2 (Remarks to the Author):

This is an interesting and potentially important contribution to our understanding the population and evolutionary biology of bacteria and lytic (virulent) phages. The manuscript includes several innovative ideas and perspectives about the conditions under which restriction-modification, abortive infection, and CRISPR-mediated immunity to phage infection will operate to protect bacteria from succumbing to phage infection. However, we cannot recommend its publication in its current form.

The authors must consider how this theoretical study of the dynamics of phage infection of individual bacteria relates to the greater picture of the population and evolutionary biology of bacteria and lytic phage. From the population and evolutionary perspective, restriction-modification is a question of enzyme kinetics, but is outcome driven- does restriction prevent replication. From this perspective, would it be possible to discriminate between fast and slow phages by their response to the host endonuclease? Stated another way, can we identify fast and slow phages from what is known about the dynamics of phage infection? Would experimentalists see differences in parameters such as: (1) the time from adsorption of the phage to the bacteria to the initiation of the lytic cycle, the latent period, (2) the rates at which the phage adsorb to the bacteria, (3) the time from the initiating of the lytic cycle to the production of free phage, (4) whether the bacteria are killed upon release of the phage or the phage ooze out of infected cells, or (5) the number of phage particles produced by an infected bacterium, the burst size.

It may pay for the authors to consider the role of classical resistance to phages by mutations of their receptors. From the evolutionary perspective, resistance is the end-all-be-all defense mechanism. How does the immunity mechanism the bacteria utilize impact its ultimate fate with respect to the emergence of resistance?

The authors also fail to consider the ecological relevance of their models. While restriction modification is ubiquitous, or nearly so, abortive infection mechanisms are less common. Moreover, functional CRISPR systems, that is CRISPR-Cas systems capable of acquiring new spacers from infection phage are rare. Therefore, a single bacterium having all three defense systems would be highly uncommon. There is rarely a need for a bacterium to make a "decision" about which system to utilize to protect themselves from phage.

We not only endorse the use of mathematical models for studies of the population and evolutionary dynamics of bacteria and phage, but we also consider them to be essential for these considerations. However, for these models to be useful, the hypothesis generated from the analyses of the properties of these models have to be testable in experimental systems. Although we don't expect the authors of this report to do these experiments, they should present these hypotheses in a way that will be amenable to experimentalists working with bacteria and phage and suggest how these hypotheses can be tested.

Reviewer #3 (Remarks to the Author):

This manuscript takes a theoretical approach to investigate interactions between bacterial immune systems. With so many novel phage defence mechanisms being discovered, understanding interactions between fundamental immunity systems such as restriction modification (RM), abortive infection (Abi) and CRISPR-Cas, is vital to improving our understanding of how and when different immunity systems act. This work has broad relevance, ranging from fundamental microbiology to application of phage therapy. Further, it highlights key hypotheses that could be tested experimentally in the future.

By developing simple models of each of these immunity systems (RM, Abi and CRISPR-Cas), the authors were able to simulate an infection landscape which explains how these systems might be complementary to one another, determined by the rate of viral replication and cleavage. These models suggest that modulation of kinetic parameters (e.g., sensitivity of the Abi sensor kinase) might enable fine-tuning of these systems to optimise how well these three immunity systems interact to minimise the chance of successful phage infection causing cell lysis. As such, the model predicts that only phages which replicate either relatively very fast or very slow, may escape all three of these immunity systems.

This manuscript demonstrates a high standard of methodology and explains the models very clearly making it accessible for non-mathematicians. Although more thorough explanation of the model details could be included at times, e.g., parameter details, what they mean and how values were determined.

Minor comments:

1. Biological relevance of model assumptions: does the strength of the immune system response generally scale with number of phage genomes present? i.e., in both models 1+2, nucleases and Abi sensors respectively scale with phage – is there an upper limit on how upregulated these systems can be?
2. The chronic infection result was interesting and counter-intuitive (Figure 2A, panel 3). Why is the immune response downregulated when there are still viral genomes present? Is there a lower limit in the model on the number of viral genomes required to trigger response?

We thank the Reviewers for their positive and constructive comments.

We modified the manuscript according to the Reviewers' suggestions, as detailed below (in blue).

Reviewer #1 (Remarks to the Author):

Arias et al. propose an interesting, and likely useful, formalism for understanding the outcome of phage infection in a cell with multiple defense systems. In their view, the defense strategy the host uses will ultimately depend on the replication rate of the phages inside the cell, and different regulatory circuits and outcomes for different types of defense necessarily lead to these distinct outcomes. They make this formalism explicit with a series of mathematical models of progressing complexity.

We thank the Reviewer for his/her positive comment on our work, in particular about the usefulness of our formalism. We have modified the text according to the indicated suggestions, as detailed below.

I have no issue with the construction of the models the authors present, which is very clear and intuitive, but ultimately the simplicity of the formalism seriously constrains its usefulness in making inferences about bacteria-phage ecological and coevolutionary dynamics.

We agree with the Reviewer that our models have a limited capacity to make inferences about ecological and evolutionary dynamics. However, this is not the aim of these models. Instead, they are intended to shed light on how anti-phage mechanisms operate at the scale of individual bacterial cells. In a way, this is more a physiological issue than an ecological or evolutionary one, since it refers to events that take place inside bacterial cells at the time-scale of the infection of a single cell. The inferences that can be made from these events at ecological and evolutionary scales are a different problem.

We have made changes in the manuscript to make our cell-level approach more explicit. We have replaced the terms "bacteria" with "bacterial cell" and "dynamics" with "intracellular dynamics" or "within-cell dynamics" throughout the text (including the title). We have also added a few lines in the Introduction to make this point clear:

"In this work, we take a cell-level perspective. The rationale for this approach is that key phage/bacteria interactions take place in the cytoplasm of infected bacterial cells. It is the outcome of these interactions that determines whether a bacterial cell undergoes suicide or creates new CRISPR spacers against an infecting phage. Therefore, understanding how bacterial immunity operates inside individual cells can provide valuable insight into relevant aspects of phage infections. To address this issue, we use simple mathematical models of the molecular mechanisms that implement anti-phage responses inside individual bacterial cells." (lines 55-61).

Overall, this manuscript reads much more like a "Perspectives" type article than a research article and may be more appropriate for submission under that category.

This seems to be an editorial remark.

Major Comments:

1. These within-cell dynamics are never embedded in a population model, seriously restricting inferences about how these defense systems will interact in even a simple 1 host – 1 phage setting. This absence is felt particularly in conversations of Abi, given that Abi is a defense that only pays a fitness payoff at a population scale.

Embedding the individual-cell dynamics in a population-scale model, as suggested by the Reviewer, would imply the formulation of new models to deal with different problems, not the reformulation of the cell-level models or any change in the results presented in this work. It is clear that the suicide induced by Abi systems in bacterial cells has important implications for the spread of the infection across the population. However, this work focuses on the causes underlying the suicide of individual cells and not on its consequences.

We have included in the Discussion a possible strategy to use the models presented here in a population-level setting:

“The infection space, as defined in this work, is a cell-scale feature: it determines the possible fates of an infected bacterial cell (i.e. whether or not it survives the infection) as a function of the rates of phage DNA replication and destruction. However, this concept can also be formalized at the scale of bacterial populations. Considering the population as a collection of individual cells, it can be characterized by the set of its infection spaces. Variations in the metabolic status or in the anti-phages implemented in each cell imply that the individual spaces within a population are not necessarily identical. The heterogeneity of the population in terms of its infection spaces is also affected by ecological and evolutionary events taking place at different time scales. This diversity plays a key role in the dynamics of the population during phage infections. For instance, some phages might be fast to some cells of the population but susceptible to others, which would entail differential mortality rates among the cell-level configurations of the infection space coexisting in the population. In turn, this would give rise to dynamic changes in the relative abundance of infection spaces in the course of phage infections. This view of the dynamics of bacterial populations as emerging from the structure of individual infection spaces suggests a promising bottom-up strategy to link intracellular and ecological aspects of phage/bacteria interactions..” (lines 361-373).

In our opinion, developing this line of research is a matter of future work.

2. Many evolutionary arguments are made throughout the paper, but the division of state space between immune systems seen is entirely a result of the assumed dynamics of different kinds of defense, not an evolved partitioning of that space.

There were only two evolutionary arguments in our manuscript:

- The first one referred to the possible selective pressures imposed on the phage DNA proliferation rate by the distribution of RM, Abi, and CRISPR systems in the infection space. We have moved this argument from the section “The CRISPR system: a bacterial strategy against slow phages” to the Discussion (see details in the response to the next point).

- The second one concerns the Abi systems: we suggest that the particular shape of the region occupied by this immune mechanism in the infection space could be modulated by natural selection. We have reformulated this argument to specify that this is a possible implication of our models and not one of their hypotheses:

“[minimizing the overlap between Abi and RM systems] could be achieved by adjusting the kinetic parameters of the sensor enzymes used by Abi systems (Fig. 3.C). This result suggests that natural selection could modulate those parameters to fine-tune the configuration of Abi systems in bacterial cells, which would determine, for instance, what phages should trigger the cell suicide depending on the ecology of each bacterial species.” (lines 195-198).

In any case, the division of the infection space between immune systems is never presented in the manuscript as an evolutionary argument. It is entirely a result of the assumed dynamics and it does not depend on any evolutionary hypothesis.

To make any evolutionary arguments the authors should formalize those arguments mathematically (e.g., by looking at the fitnesses of competing strains in the model, perhaps taking a game-theoretic approach)

We agree with the Reviewer that the arguments about the selective pressure imposed on the DNA proliferation rate by anti-phage defenses require a more appropriate formalization. We have moved these arguments to the discussion section. We have included an explanation of how the concept of infection space (a cell-level feature) can be generalized to the scale of the population :

“The concept of an infection space also admits an alternative, static formalization at a population scale. From a population viewpoint, susceptible phages can be defined as those that are susceptible to all the cells of the population. From this definition, the region of the population-level infection space occupied by susceptible phages is given by the intersection of the corresponding regions in the cell-level infection spaces contained in the population. An analogous argument can be used to define phages as slow or fast at the scale of the population. The heterogeneity of individual-level infection spaces implies a greater complexity for the population-level space since new regions appear where phages may be susceptible to some bacterial cells but not to others.” (lines 374-380).

Using this formalization we highlight the importance of the rate of DNA replication in the interactions between phages and bacteria, which could make it a potential target of natural selection:

“Even a simple setting of the population-level space allows exploring interesting evolutionary consequences of immune mechanisms that operate within the cytoplasm of individual bacterial cells. For instance, it is clear that phages should avoid the regions of the population-level infection space where they can be neutralized by all the cells of the population (such phages would be unable to complete their life cycle and progress with the infection). A possible phage strategy to achieve that end, widely documented in the literature [47, 55], consists in modifying the restriction sites or the sequences targeted by CRISPR

systems in the viral DNA, which reduces the likelihood of nuclease-mediated recognition and cleavage and, consequently, lowers the rate of phage destruction. Within the framework of the infection space, this strategy would allow phages to escape from the susceptible regions. Our models suggest the existence of a less obvious selective pressure on phages. By increasing or decreasing their rates of proliferation inside their bacterial hosts, phages could also avoid immune destruction by evading the action of bacterial nucleases. From this viewpoint, the Abi and CRISPR systems could be considered as adaptations to counteract these phage strategies.” (lines 381-391)

3. The argument made in the section “How do bacteria decide to include new phages in the CRISPR array?” is entirely verbal, and has no mathematical analysis. I suggest the authors formalize this intuition in equations. This entire section is largely speculation. In particular, the last paragraph of this section makes a number of claims, none of which I think are actually addressed by the model the authors present (the decision to incorporate a spacer and what mechanisms make that decision, why CRISPR systems would target slow phage – see next section)

The arguments used in this section are by no means verbal. They are deduced from the results of our models. However, we acknowledge that the presentation of these arguments might not be sufficiently clear in the manuscript. We have made extensive changes in this section (including the legend of Fig. 6) to facilitate its comprehension (see lines 258 through 308). We hope that the logic of our arguments and how they are deduced from the models is more clear now.

4. The treatment of CRISPR systems in the paper, while a central point of the work (see title), is extremely simplistic, and I would argue not particularly helpful in terms of understanding these systems. CRISPR systems are implemented in Model 3 identically to an RM system with slightly different parameters – in fact one could think of the same model as representing a cell with an Abi system and two RM systems with different regulatory and kinetic parameters. Nothing is said, really, about adaptive immunity in these models.

This point might not be sufficiently explained in the original text. Model 3 is intended to simulate the effects of CRISPR systems on the dynamics of phage infections, in particular on the degradation of the phage DNA. This model is used to simulate a scenario in which the infected bacterial cell already has CRISPR spacers that target the infecting phage. In this case, the behavior of the Cas nucleases is not different from that of restriction nucleases: they are upregulated after the detection of the infection, they interact with the phage DNA, and finally, they are downregulated. As the Reviewer points out, in this situation there are two sets of enzymes that degrade the phage DNA with different regulatory and kinetic parameters. Other aspects of adaptive immunity (such as the adaptation or the expression phases) are not relevant in this context. We acknowledge that this point might be sufficiently explained in the text. We have reformulated the goals of Model 3 to make these arguments clear:

“To understand the protection conferred on bacterial cells by the CRISPR system, we will model the dynamics of this system in the cytoplasm of infected cells. Since we intend to analyze the effect of CRISPR on the dynamics of the phage DNA inside the host cell, we will not consider the adaptation and expression phases of the CRISPR response. Instead, we

will assume that the infected bacterial cell is already equipped with spacers that recognize and attack the DNA of the phage. In this case, the infection activates the interference stage of the CRISPR response in the host cell, which is the focus of our model.“ (lines 219-223)

Also, importantly, there is no real justification for why CRISPR systems would have parameters that would lead them to better target slow phage.

That CRISPR systems could target slow phages is a hypothesis of our work. We have changed the first lines of the section “The CRISPR system: a bacterial strategy against slow phages” to better explain this point:

“We have seen that, according to our models, RM and Abi systems would fail to protect bacterial cells against slow phages (Fig. 3). Based on this, we hypothesize that CRISPR systems could be optimized to fight this category of phages. Under this hypothesis, the adaptive memory of bacterial cells would target phages that are not susceptible to the action of RM and Abi systems. Innate and adaptive immunity would therefore play complementary roles in the defense of bacterial cells against infecting phages. This hypothesis is supported by two main arguments. First, we will show that adaptive immune responses protect bacterial cells from slow phage infections. Second, we will show that bacterial cells could be able to discriminate susceptible and fast phages from slow phages and selectively incorporate the latter into their CRISPR array“ (lines 211-217)

We use mathematical models to show that our hypothesis is plausible, i. e. that CRISPR systems can be effective against slow phages. Model 3 shows that this is the case: bacterial cells that have CRISPR spacers can neutralize phages that are resistant to RM systems. We do not say that CRISPR systems have parameters that lead them to better target slow phages. We show that this is a possible scenario, i.e that suitable parameter values can be selected so that solutions of the model display the behaviors described in the text. We have changed the legend of Fig. 4 to make the use of Model 3 clear.

The last paragraph of this section makes a number of evolutionary arguments that are not backed up or formalized at all – and may not hold at all in a population setting.

We have moved these evolutionary arguments to the Discussion (the details are explained in the response to point 2 above).

5. The priming phenomenon common to most CRISPR systems (rapid uptake of spacers against phages the host already has immunity towards) is not mentioned in the paper, and seriously complicates the arguments about spacer acquisition decisions made by the authors (though see <https://doi.org/10.1073/pnas.1602639113> for information that may be useful/supportive of the argument). Similarly, while the authors correctly point out that spacers can be acquired from the byproducts of RM defense, defective phage can also facilitate spacer uptake (<https://doi.org/10.1038/ncomms5399>).

We thank the reviewer for these (quite interesting) bibliographic references. The first one suggests that the decision of bacterial cells to create new spacers during reinfections might be determined by the permanence of phage DNA in their cytoplasm. This is a relevant

element of our hypothesis about the creation of new spacers. We have included this reference to support this argument:

“We hypothesize that the incorporation of new spacers into the CRISPR array of a bacterial cell is determined by the permanence of phage DNA in its cytoplasm (as already suggested in [47]). (lines 249-251)

This paper also supports the assumption that the creation of new spacers during reinfections would be facilitated by a reduction in the rate of phage DNA destruction. We have included this argument in the section “How do bacteria decide to include new phages in the CRISPR array?”:

“This model would account for the increased acquisition of new spacers during reinfections by phages that have undergone mutations in the targeted DNA sequences as compared to phages without mutations [47]. According to our model, this would be the expected result of lower cleavage rates caused by the reduced affinity of Cas enzymes for the phage DNA.” (lines 305-308)

The second paper proposed by the Reviewer supports the view of the CRISPR system as a strategy to fight slow phages. The reduced replication rate of defective phages would make them slower than normal phages, and consequently, suitable targets for the CRISPR systems:

“This model is supported by the observation that lower replication rates caused by the presence of replication-defective phages facilitate the acquisition of new spacers by bacterial cells during phage infections [48].” (lines 262-266)

6. The CRISPR modeling literature is extremely rich, grapples with many of the questions discussed here, and largely ignored in this manuscript (with a few exceptions). For example, <https://doi.org/10.1371/journal.pcbi.1004603> seems particularly relevant.

We thank the Reviewer for this reference. We have included it in the Introduction (together with other references) to put our work in context and to stress our cell-level approach:

“These questions have been addressed at ecological and evolutionary scales in the literature [33–35]. In this work, we take a cell-level perspective.” (lines 55-56).

7. There is no exploration of parameter space nor justification of the parameters used in the manuscript, making it hard to judge the generality of the results shown.

The choice of parameters in Model 1 corresponds to a non-dimensional form of its equations. The simplicity of this model allows exploring its dynamics by changing two parameters: the rates of phage DNA replication and cleavage. If the rest of the conditions are equal, phages that differ only in their rates of replication and destruction will exhibit differences in their susceptibility to the host’s immune defenses. Therefore, the infection space represents the exploration of the parameter space. We have added this explanation to the text:

“A non-dimensional version of Model 1 allows exploring all the dynamics produced by this model by varying two parameters: α and β (see Material and Methods). Using this approach, it is possible to define an “infection space” in which the outcome of anti-phage responses of individual bacterial cells is fully characterized by the rates of phage DNA replication (represented by α) and degradation by restriction nucleases (represented by β).” (lines 120-123)

We have also included the non-dimensional version of Model 1 in the Materials and Methods section (lines 408-412).

With regard to the Abi systems, we intended to show that they can target fast phages. To that end, it suffices to find parameter values in Model 2 that make the Abi region of the infection space overlap with that of fast phages. Analogously, we have selected suitable parameters to show that the CRISPR systems can be effective against slow phages (see details in the response to point 4).

We have explained these points in the Materials and Methods section:

“For Model 2 and Model 3 we chose suitable values to show that Abi and CRISPR systems can be effective against fast and slow phages respectively.” (lines 411-412).

Minor Comments:

1. Please make all code for analyses publicly available (e.g., on github)

We made the code available at <https://notebookarchive.org/the-coordination-of-innate-and-adaptive-immunity-in-bacterial-cells--2022-08-dbzqzpq/>. We added a Code availability section at the end of the manuscript.

2. It seems like many of these models should be amenable to mathematical analysis, not just numerical approaches, and such solutions would be quite useful

Numerical approaches are in fact mathematical analysis. If the Reviewer refers to explicit solutions instead, they are (to the best of our knowledge) currently unavailable.

3. The line “This selective force drives frequent mutations...” is incorrect – depletion of restriction sites does not drive mutation. The word “mutations” should be replaced by “substitutions”

We thank the Reviewer for this observation. We have changed the text accordingly.

Reviewer #2 (Remarks to the Author):

This is an interesting and potentially important contribution to our understanding the population and evolutionary biology of bacteria and lytic (virulent) phages. The manuscript includes several innovative ideas and perspectives about the conditions under which restriction-modification, abortive infection, and CRISPR-mediated immunity to phage infection will operate to protect bacteria from succumbing to phage infection. However, we cannot recommend its publication in its current form.

We thank the Reviewer for his/her positive comments. We have modified the text according to the indicated suggestions, as detailed below.

The authors must consider how this theoretical study of the dynamics of phage infection of individual bacteria relates to the greater picture of the population and evolutionary biology of bacteria and lytic phage.

We have included in the text two possible ways to formalize the concept of a cell-level infection space at the scale of the population and use it to explore the ecological and evolutionary implications of events that occur inside bacterial cells (lines 361-396).

We have also included a remark about the population-level manifestations of the infection of single cells by fast and slow phages (see the response to the next point).

From the population and evolutionary perspective, restriction-modification is a question of enzyme kinetics, but is outcome driven- does restriction prevent replication. From this perspective, would it be possible to discriminate between fast and slow phages by their response to the host endonuclease? Stated another way, can we identify fast and slow phages from what is known about the dynamics of phage infection?

This is a very interesting point. Phages are defined as fast or slow depending on the strategy they use to evade intracellular immunity. Although both strategies may kill individual host cells, they may exhibit very different population-level consequences, which would facilitate the experimental discrimination between fast and slow infections. Although we were not aware of this, the existence of persistent phage infections has already been described in the literature (see DOI: [10.1098/rsob.130200](https://doi.org/10.1098/rsob.130200) or DOI: [10.1016/B978-0-12-800259-9.00004-4](https://doi.org/10.1016/B978-0-12-800259-9.00004-4)). According to our models, these infections would be caused by slow phages. We have included these arguments in the manuscript:

“Being fast or slow would be alternative phage strategies to evade the immune mechanisms of individual bacterial cells. Although both strategies may lead to the death of the infected cell, their consequences at the scale of bacterial populations would likely be different. Fast phages are capable of rapidly killing their hosts, which probably translates into the high transmission and mortality rates observed in typical lytic infections. The chronic infection of individual cells by slow phages, on the other hand, would result in persistent infections with lower death rates at the scale of the population. This type of infection has been observed in natural populations in which bacteria and phages coexist in a more or less stable equilibrium known as the carrier state life cycle [43]. This state is characterized by persistent infections in which new phages are continuously budded off the host cells or passed down to the

progeny of the infected bacterial cells [42]. Within the framework of the infection space, this would be the population-level manifestation of infections by slow phages.” (lines 129-138)

Would experimentalists see differences in parameters such as: (1) the time from adsorption of the phage to the bacteria to the initiation of the lytic cycle, the latent period, (2) the rates at which the phage adsorb to the bacteria, (3) the time from the initiating of the lytic cycle to the production of free phage, (4) whether the bacteria are killed upon release of the phage or the phage ooze out of infected cells, or (5) the number of phage particles produced by an infected bacterium, the burst size.

Some of these parameters correspond to variables of our models but others refer to processes that are not considered in this work. Let us briefly address this issue below.

1. The latent period, defined as “the timing of phage-induced host cell lysis” (<https://doi.org/10.1128/AEM.67.9.4233-4241.2001>), is an output of our models. This type of temporal output is shown for instance in Figures 6A (for restriction-modification systems without CRISPR) and 6D (with CRISPR). This figure shows the time needed by the cell to eliminate the phage but the time needed for the phage to kill the host is a similar output. Since it is a result of our models, the latent period can be expressed as a function of other model parameters such as the phage DNA replication and destruction rates. Experimentally changing these parameters should give rise to predictable changes in the latent period.

2. The absorption of phages into bacteria takes place before the infection of the cell. Since our models refer to events that take place once the phage has entered the host cell, they cannot say anything about the rates at which the phages adsorb to the bacterial cells. This parameter would be useful in the population-level implementation of our models.

3. Our models do not explicitly simulate the production of new phages, so they cannot provide valuable information about this variable.

4. As discussed in the response to the previous point, the death of the bacteria upon release of new phages and the continuous release of phages out of infected cells could respond to fast and slow phage strategies respectively. According to our models, fast phages would give rise to typical lytic cycles, whereas slow phages would create chronic infections leading to carrier state life cycles.

5. The number of phage particles produced by an infected bacterium (the burst size) affects the rate of propagation of the infection across the population, so it would be useful in population-level versions of our models.

It may pay for the authors to consider the role of classical resistance to phages by mutations of their receptors. From the evolutionary perspective, resistance is the end-all-be-all defense mechanism. How does the immunity mechanism the bacteria utilize impact its ultimate fate with respect to the emergence of resistance?

This question is interesting from an evolutionary perspective. However, our models focus on the interactions of bacterial cells with phages that are capable of infecting them and are not intended to address this particular aspect of the coevolution of phages and bacteria.

The authors also fail to consider the ecological relevance of their models. While restriction modification is ubiquitous, or nearly so, abortive infection mechanisms are less common. Moreover, functional CRISPR systems, that is CRISPR-Cas systems capable of acquiring new spacers from infection phage are rare. Therefore, a single bacterium having all three defense systems would be highly uncommon. There is rarely a need for a bacterium to make a “decision” about which system to utilize to protect themselves from phage.

The simultaneous presence of all the defense systems in every bacteria is not an assumption of our approach. Our models explain why some phages evade specific immune mechanisms and also how bacterial cells with more than one immune system “decide” which system to utilize. According to the literature, the CRISPR system is present in 40% of bacterial species (reference 19 in our manuscript, doi: 10.1016/j.molcel.2009.12.033). Since restriction modification is ubiquitous, bacteria with at least two anti-phage mechanisms are not rare at all. We have included a comment in the Discussion about this issue:

“The configuration of the infection space presented in this work does not intend to be an accurate, universal description of bacterial infection spaces. For one thing, it corresponds to bacterial cells equipped with RM, Abi, and CRISPR systems, a situation that is not the norm among bacterial species. Moreover, the shape of the regions occupied by the Abi, and CRISPR responses in this space depends on the particular mathematical formalization used to model their underlying molecular mechanisms. Instead, the infection space is intended as a useful abstraction of phage/bacteria interactions whose main utility derives from its capacity to provide valuable insight into the within-cell dynamics of phage infections. The interest of this approach transcends the particular situations considered in our models. For instance, bacterial cells that only use RM and Abi systems would still be susceptible to slow phages, whereas the combination of RM and CRISPR responses alone would not suffice to protect the host against fast phages.” (lines 352-360)

We not only endorse the use of mathematical models for studies of the population and evolutionary dynamics of bacteria and phage, but we also consider them to be essential for these considerations. However, for these models to be useful, the hypothesis generated from the analyses of the properties of these models have to be testable in experimental systems. Although we don't expect the authors of this report to do these experiments, they should present these hypotheses in a way that will be amenable to experimentalists working with bacteria and phage and suggest how these hypotheses can be tested.

We thank the Reviewer for this suggestion. We have included a list of the hypotheses in the Discussion:

“[...] suggests testable hypotheses about the coordination of immune mechanisms in individual bacterial cells:

- Phages with high and low DNA replication rates could avoid the action of restriction nucleases in the cytoplasm of infected cells. We have labeled these phages as fast and slow respectively. Whereas fast phages would give rise to typical lytic cycles, slow phages would lead to the persistent infections observed in carrier state life cycles.
- The Abi and CRISPR systems would be optimized to target fast and slow phages respectively.

- The decision of bacterial cells to create new spacers would be determined by the permanence of the phage DNA in the cytoplasm of the cell and the timing of activation of the Cas enzymes. As a consequence, the creation of new spacers in the CRISPR array would be biased towards slow phages.
- During reinfections, bacterial cells would create new spacers against phages that are already present in their CRISPR library if the rate of phage DNA destruction has decreased since previous infections (owing to modifications in the restriction sites recognized by nucleases or to mutations in the sequences targeted by the spacers present in the CRISPR array of the host cell). (lines 318-329)

We have also outlined some possible empirical approaches to this issue:

“The mathematical models presented in this work support the plausibility of these hypotheses but their validation requires an empirical approach. In this regard, techniques that allow monitoring single-cell dynamics [51] could be used to experimentally test if different infection strategies (e.g., lytic or chronic) respond to different rates of phage DNA replication and cleavage. Alternatively, manipulating those rates would lead to predictable changes in the outcome of phage infections. For instance, it would be theoretically possible to shift between the carrier state and the lytic life cycles by artificially decreasing or increasing the rate at which the DNA of a given phage replicates inside bacterial cells.” (lines 330-336)

Reviewer #3 (Remarks to the Author):

This manuscript takes a theoretical approach to investigate interactions between bacterial immune systems. With so many novel phage defence mechanisms being discovered, understanding interactions between fundamental immunity systems such as restriction modification (RM), abortive infection (Abi) and CRISPR-Cas, is vital to improving our understanding of how and when different immunity systems act. This work has broad relevance, ranging from fundamental microbiology to application of phage therapy. Further, it highlights key hypotheses that could be tested experimentally in the future.

We thank the Reviewer for his/her positive remarks about our manuscript. We have modified the text according to the indicated suggestions, as detailed below.

By developing simple models of each of these immunity systems (RM, Abi and CRISPR-Cas), the authors were able to simulate an infection landscape which explains how these systems might be complementary to one another, determined by the rate of viral replication and cleavage. These models suggest that modulation of kinetic parameters (e.g., sensitivity of the Abi sensor kinase) might enable fine-tuning of these systems to optimise how well these three immunity systems interact to minimise the chance of successful phage infection causing cell lysis. As such, the model predicts that only phages which replicate either relatively very fast or very slow, may escape all three of these immunity systems.

This manuscript demonstrates a high standard of methodology and explains the models very clearly making it accessible for non-mathematicians. Although more thorough explanation of the model details could be included at times, e.g., parameter details, what they mean and how values were determined.

We have included an explanation of the meaning of the parameters after the formulation of each model.

- Equations 1:

“where $g(t)$ and $n(t)$ are the numbers of phage genomes and of nucleases in the cytoplasm of the host cell at time t respectively. The infection of the bacterial cell is assumed to start at time $t = 0$ with the entry in its cytoplasm of g_0 phage genomes. Throughout this text, we further assume that there are no superinfections, i.e. that phages prevent the entry of other phages into the host cell once the infection has started [38]. Parameter g_{\min} represents the lower threshold that determines phage viability and g_{\max} the number of phage genomes that causes the lysis of the host cell.” (lines 89-93)

- Equations 2:

“ λ and μ are positive parameters that represent the resistance of nucleases to expand and the force exerted by the phage DNA on the number of bacterial nucleases respectively [40].” (lines 108-110)

- Equations 3:

“where ϕ , γ , and δ are positive parameters. This model simulates the dynamics of an intracellular protein that is normally produced at a rate ϕ and disappears exponentially at a rate γ . These dynamics result in a homeostatic equilibrium given by $s = \phi/\gamma$, which is taken

as the initial condition for s . The presence of the phage in the bacterial cytoplasm induces an additional loss of the protein, which is assumed proportional to the amount of phage DNA (parameter δ is the constant of proportionality).” (lines 153-156)

- Equations 4:

“where $s(t)$ and $g(t)$ are the numbers of sensor proteins and phage genomes in the cytoplasm of an infected bacteria cell at time t respectively. Positive parameters γ and δ represent the rate at which protein s is produced in response to the phage DNA and the rate of protein degradation respectively. This model assumes that the concentration of the sensor protein s in an infected bacterial cell increases with the number of genomes in its cytoplasm and decreases exponentially.” (lines 183-187)

Regarding the values of the parameters, we have used a non-dimensional form of Model 1 that allows exploring the dynamics of this model by playing with just two parameters: the rates of phage DNA replication and cleavage. If the rest of the conditions are equal, phages that differ only in their rates of replication and destruction will exhibit differences in their susceptibility to the host’s immune defenses.

We have included the non-dimensional version of Model 1 in the Materials and Methods section (lines 408-410).

With regard to the Abi systems, we have made a suitable choice of the parameters in Model 2. In this case, we were interested in showing that the Abi systems can target fast phages. To that end, it suffices with finding parameter values verifying this condition. Analogously, we have selected suitable parameters to show that the CRISPR systems can be effective against slow phages.

We have added an explanation of these points in the Materials and Methods section:

“For Model 2 and Model 3 we chose suitable values to show that Abi and CRISPR systems can be effective against fast and slow phages respectively.” (lines 411-412).

Minor comments:

1. Biological relevance of model assumptions: does the strength of the immune system response generally scale with number of phage genomes present? i.e., in both models 1+2, nucleases and Abi sensors respectively scale with phage – is there an upper limit on how upregulated these systems can be?

This is an interesting question. We have not found any empirical evidence to support the existence of this limit but it could well exist. This detail could be easily included in our models (for instance by assuming an upper boundary to the number of nucleases), which could probably change the particular configuration of the infection spaces shown in the Figures of the manuscript. However, the qualitative features of this space (and consequently the conclusions that can be deduced from it) would likely remain similar.

2. The chronic infection result was interesting and counter-intuitive (Figure 2A, panel 3).

Although we were not aware of it, this type of infection has actually been observed in natural populations. We have included a few lines about this in the text:

“Being fast or slow would be alternative phage strategies to evade the immune mechanisms of individual bacterial cells. Although both strategies may lead to the death of the infected cell, their consequences at the scale of bacterial populations would likely be different. Fast phages are capable of rapidly killing their hosts, which probably translates into the high transmission and mortality rates observed in typical lytic infections. The chronic infection of individual cells by slow phages, on the other hand, would result in persistent infections with lower death rates at the scale of the population. This type of infection has been observed in natural populations in which bacteria and phages coexist in a more or less stable equilibrium known as the carrier state life cycle [42]. This state is characterized by persistent infections in which new phages are continuously budded off the host cells or passed down to the progeny of the infected bacterial cells [43]. Within the framework of the infection space, this would be the population-level manifestation of infections by slow phages..” (lines 129-138)

Why is the immune response downregulated when there are still viral genomes present? Is there a lower limit in the model on the number of viral genomes required to trigger response?

The number of viral genomes acts in Model 1 as a “force” that fosters the upregulation of nucleases. Since nucleases are modeled as an elastic system, they are subject to an opposite, elastic force, that tends to bring the system back to its original state. This is what allows this model to simulate both the initial upregulation of nucleases and their eventual downregulation. Under some circumstances, the force exerted by the phage genomes may not be sufficient to compensate for the tendency of the system to recover its initial state, hence the downregulation of the response before the disappearance of the viral genomes from the cytoplasm. Therefore, this behavior does not depend on the number of genomes falling below a certain limit. This limit does not emerge in the model, since numerical simulations with even a very small number of genomes will lead to a (possibly very small) increase in the number of nucleases.

Reviewer #1 (Remarks to the Author):

The authors have done a good job addressing most of my comments.

I am still concerned, though, about their treatment of CRISPR systems. Their title makes a claim to say something specific about adaptive immunity, yet their treatment of CRISPR is, as far as I can tell, mathematically identical to an RM system but with different parameters included. I understand the authors mean this to be a "hypothesized" parameter space for CRISPR, but I do not see good justification or links to why CRISPR's adaptive nature is important to their arguments. I think, as a core point of the paper, this needs to be made clear from the outset.

With respect to the population/evolutionary implications of the paper - both I and another reviewer pointed these out as important to actually support the claims the authors make (especially because CRISPR and abi systems have important behaviors/payoffs only seen at the population level). The authors chose to sidestep these comments, which is a little disappointing, though I still think the paper has merit in its current form. While the authors claim that only two evolutionary arguments are made in their manuscript, language discussing "optimization" and optimal strategies are sprinkled throughout, suggesting otherwise.

Reviewer #3 (Remarks to the Author):

The revisions and responses the authors have provided fully satisfy my previous comments.

Reviewer #1 (Remarks to the Author):

The authors have done a good job addressing most of my comments.

We thank the Reviewer for his/her valuable comments, which have greatly contributed to improve the final version of this article.

I am still concerned, though, about their treatment of CRISPR systems. Their title makes a claim to say something specific about adaptive immunity, yet their treatment of CRISPR is, as far as I can tell, mathematically identical to an RM system but with different parameters included. I understand the authors mean this to be a "hypothesized" parameter space for CRISPR, but I do not see good justification or links to why CRISPR's adaptive nature is important to their arguments. I think, as a core point of the paper, this needs to be made clear from the outset.

We agree that the adaptive nature of the CRISPR system is not necessary to support our arguments. We have removed the references to adaptive aspects of bacterial immunity in the text (including the title). Regarding the action of Cas nucleases during CRISPR responses, it is analogous to that of RM systems in that it exhibits an elastic nature, i.e. Cas nucleases are upregulated and downregulated in the course of a phage infection. This is the rationale for using the same modeling approach in both cases. We have tried to make this point clearer in the text:

"As occurs with RM systems, Cas nucleases must expand after the detection of a pathogen, and contract once the phage is eliminated. Therefore, they can also be viewed as elastic systems, which implies that the dynamics of Cas nucleases are conceptually identical to restriction nucleases. For this reason, we will assume that they respond to the same model as RM systems, possibly with different parameters values. Under this assumption, the within-cell dynamics of Cas nucleases can be described by the following equations"

With respect to the population/evolutionary implications of the paper - both I and another reviewer pointed these out as important to actually support the claims the authors make (especially because CRISPR and abi systems have important behaviors/payoffs only seen at the population level). The authors chose to sidestep these comments, which is a little disappointing, though I still think the paper has merit in its current form.

It was not our intention to sidestep the evolutionary and ecological questions raised by the Reviewers. The models proposed in this work, as well as their consequences, do not rely on evolutionary arguments. They focus on anti-phage mechanisms present in bacterial cells and on they are coordinated during infections, regardless of their evolutionary origin. However, we agree that these models have interesting evolutionary and ecological consequences. We have considered two possible ways in which these consequences could be explored but we believe that this is a matter for future work.

While the authors claim that only two evolutionary arguments are made in their manuscript, language discussing "optimization" and optimal strategies are sprinkled throughout, suggesting otherwise.

We did not intend any evolutionary connotation in the use words such as "optimal strategies". We have changed these expressions to make this more clear.

Reviewer #3 (Remarks to the Author):

The revisions and responses the authors have provided fully satisfy my previous comments.

We thank the Reviewer for his/her positive evaluation of this work.